behaviour/ecology

mass mortality, carcass decomposition, fear ecology, scavengers, *Corvus corvax*, *Rangifer tarandus*

**Author for correspondence:**
S. C. Frank
e-mail: shane.frank@usn.no

# Fear the reaper: ungulate carcasses may generate an ephemeral landscape of fear for rodents

S. C. Frank[1], R. Blaalid[2], M. Mayer[3], A. Zedrosser[1,4] and S. M. J. G. Steyaert[1,5]

[1]Faculty of Technology, Natural Sciences and Maritime Sciences, Department of Natural Sciences and Environmental Health, University of South-Eastern Norway, 3800 Bø i Telemark, Norway
[2]Norwegian Institute for Nature Research, Thormøhlensgate 55, 5006 Bergen
[3]Department of Bioscience, Aarhus University, 8410 Rønde, Denmark
[4]Department of Integrative Biology, Institute of Wildlife Biology and Game Management, University of Natural Resources and Applied Life Sciences, Vienna, Austria
[5]Faculty of Biosciences and Aquaculture, Nord University, 7711 Steinkjer, Norway

SCF, 0000-0001-8153-6656; RB, 0000-0002-3883-8189; MM, 0000-0002-9905-3625; AZ, 0000-0003-4417-3037; SMJGS, 0000-0001-6564-6361

Animal carcasses provide an ephemeral pulse of nutrients for scavengers that use them. Carcass sites can increase species interactions and/or ephemeral, localized landscapes of fear for prey within the vicinity. Few studies have applied the landscape of fear to carcasses. Here, we use a mass die-off of reindeer caused by lightning in Norway to test whether rodents avoided larger scavengers (e.g. corvids and fox). We used the presence and abundance of faeces as a proxy for carcass use over the course of 2 years and found that rodents showed the strongest avoidance towards changes in raven abundance ($\beta = -0.469$, s.e. = 0.231, $p$-value = 0.0429), but not fox, presumably due to greater predation risk imposed by large droves of raven. Moreover, the emergence of rodent occurrence within the carcass area corresponded well with the disappearance of raven during the second year of the study. We suggest that carcasses have the potential to shape the landscape of fear for prey, but that the overall effects of carcasses on individual fitness and populations of species ultimately depend on the carcass regime, e.g. carcass size, count, and areal extent, frequency and the scavenger guild. We discuss conservation implications and how carcass provisioning and landscapes of fear could be potentially used to manage populations and ecosystems, but that there is a gap in understanding that must first be bridged.

# 1. Introduction

Animal carcasses and the scavengers that use them are important for ecosystem processes and structure [1–3]. These processes are underpinned by ecological and species interactions [4–6]. Animals take advantage of carrion either through the direct consumption of detritus or indirectly by feeding on arthropod 'blooms' in and around carcass sites [7–9].

Carcasses can attract a host of both obligate and facultative scavengers (hereafter scavengers), creating a localized increase in species occurrence, their interactions, and predation risk for smaller and more vulnerable species [10]. Such locales can be considered 'islands of risk' or 'hills' for smaller and/or more vulnerable species, whether scavengers or non-scavengers, in the vicinity [11–13]. Consequently, these risky areas become a part of the 'landscape of fear' (LOF; [14]), i.e. defined as the spatially explicit distribution of perceived predation risk as seen by prey [15].

The LOF framework has provided a better understanding of animal decisions in relation to food and safety trade-offs, predator–prey relationships and how communities are structured across trophic levels [3,16,17]. Despite the current popularity of the LOF and the occurrence of increased predation risk within the vicinity of carcasses [18–20], few studies have applied the LOF concept to carcass sites (e.g. 12,21).

Moreover, the level of fear induced at a carcass site could be related to its attractiveness or the amount of carcass biomass [13]. The most extreme case of carcass biomass, a mass mortality event (MME; e.g. greater than or equal to 10 individuals for mammals in [22]), can have long-lasting effects on populations, and alter patterns in ecological interactions and biodiversity [23–27]. These effects are likely influenced by the amount and distribution of carrion from an MME [28] and mediated by species who use them [20,29]. To our knowledge, no study has examined evidence for an LOF following an MME. Whether single carcasses or an MME, carrion is ephemeral in nature [18] and likely translates to a dynamic perception of risk and fear among the species who interact, use and occur near such sites [30].

We hypothesized that MMEs attract a host of species, some of which are scavengers, predators and/or prey, and that this creates a landscape of fear for prey who might also take advantage of carcass resources. We predict that prey would inversely spatially track their potential predators at an MME site. This is to say that prey will avoid areas where predators are present and appear in areas where predators become absent. Here, we evaluate evidence for a LOF at an MME site across 2 years. We used spatially referenced faeces of rodents (mostly voles) and of larger scavengers (birds and mammalian mesopredators) as a proxy for their spatial use and expected (H1) that larger scavengers' use of carcasses would decrease over time as resources depleted, i.e. soft tissues that are more attractive to raven and fox, (H2) that rodents would spatially avoid larger scavengers when present and (H3) that rodents would increase their use of the carcass area over time in conjunction with the absence of larger scavengers, because although depleted, potential carcass resources would still be available.

# 2. Methods

## 2.1. Study site

Our study site is located 1220 m.a.s.l. in an alpine ecosystem of the Hardangervidda plateau in southcentral Norway. It has mean temperatures of 10.5°C and –6.4°C for the months of July and December, respectively, and mean annual precipitation of 860 mm, with snow cover typically lasting from October until late May (Norwegian Meteorological Institute 2017), and with a small north-northeast facing slope. Here, a lightning strike killed a herd ($N = 323$) of wild tundra reindeer (*Rangifer tarandus*) on 26 August 2016. Carcasses span an area of approximately 240 × 100 m, with their highest concentration within roughly 50 × 50 m (figure 1). State officials removed all heads from the carcasses for chronic wasting disease screening, but all else remained on-site. The field- and ground layers are species poor with dwarf birch (*Betula nana*), ericaceous species, graminoids, mosses and lichens dominating.

## 2.2. Data collection

Animal species observed on the study site include corvids (raven *Corvus corax* and hooded crow *C. cornix*), golden eagle (*Aquila chrysaetos*), foxes (red fox *Vulpes vulpes* and artic fox *V. lagopus*), wolverine (*Gulo gulo*) and several rodent species (e.g. Arvicolinae) [27]. Some but not all Arvicolinae rodents scavenge on carcasses [31,32] and we were not able to discern faeces among rodent species; therefore, we refer to them collectively as 'rodents'. Rodent species in the study area include the root vole (*Microtus*

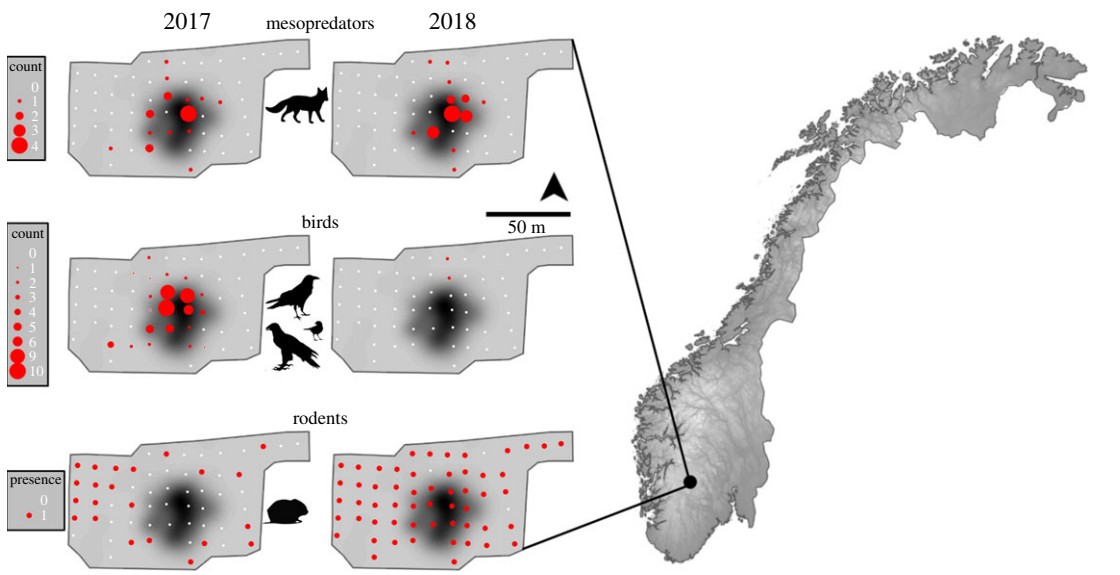

**Figure 1.** Use of ungulate carcasses (faeces abundance or the presence of a faecal pellet group) from an MME, by mesopredators, birds and rodents in 2017 and 2018. White dots are 1 × 1 m plots within our sampling grid across the study site. Abundance and presence of faeces (red) are centred on them and their size is proportional to abundance. Note that some white dots of plots are obscured by red dots of faeces abundance or presence. The background is a kernel density of the carcasses, with darker areas representing higher carcass density. The location of the study site is shown within southcentral Norway along with the elevation of surrounding areas (dark grey to light shading is low to high elevation: 0 to ∼2300 m).

*oeconomus*), lemming (*Lemmus lemmus*), bank vole (*Myodes glareolus*), field vole (*Microtus agrestis*) and the grey red-backed vole (*Myodes rufocanus*) [33]. All other species, i.e. scavengers, are hereafter referred to as 'larger scavengers', because of (i) their larger body size compared to rodents, and (ii) the mortality risk larger scavengers pose to rodents (e.g. [34]). The bird species that predominated in visits to the carcasses was raven, which occurred in large groups [27].

We set up a 10 × 10 m grid containing 59 1 × 1 m survey plots covering the study site (figure 1). Each plot was subdivided into four 50 × 50 cm quadrants in which two observers systematically searched each for 30 s, totalling 4 min per plot, and recorded the number of (i) mammalian mesopredator faeces (i.e. red and artic fox, hereafter 'mesopredators'), (ii) bird faeces, and (iii) the presence of rodent faecal pellet groups (greater than or equal to 1 faecal pellet detected). All faeces were left *in situ* for other ongoing studies. The same two observers collected these plot-level data for two consecutive years during autumn on single day visits: 11 August 2017 and 4 August 2018. In addition, detailed cryptogams that included plants identified to species or genus level for each year were conducted for another ongoing study and species were clustered into functional groups including vascular plants, graminoids, mosses and lichens. Per cent coverage for these plants and for soil, stone and carcass for the northwest quadrant of each plot were estimated visually. The mean temperatures and precipitation for the months of July and August (leading up to our visits) were very similar from 2017 to 2018 (precipitation: 2.73–2.92 mm, s.d.: 4.95–7.58; temperature: 10–10°C, s.d.: 0.756–0.802, respectively; https://www.yr.no).

## 2.3. Statistical analyses

To evaluate evidence for rodents operating within a landscape of fear in relation to the abundance of larger scavengers at the study site, we (i) used generalized linear models (GLMs) to first determine whether the use of the study site by rodents (faecal pellet presence) and larger scavengers (faecal counts) differed between years and in relation to carcass density [27], and we used (ii) multinomial logistic regression (MLRMs) to model the effect of larger scavengers' use on changes in the presence–absence of rodents over the same 2 years. For mesopredator and bird abundance response variables, we fitted each to a Poisson distribution and for rodent presence–absence we used a Bernoulli distribution. Each response for the GLMs consisted of candidate models representing a different search radius (1–10, 15, 20, 30, 40, 50, 100, 200 m) when calculating the carcass kernel density [27]. Mesopredators and birds have been shown to use the higher carcass density areas of the study site, whereas rodents avoided them, during 2017 [27]. As carcass density and distances to the nearest carcass did not change across years and was positively correlated with larger scavenger abundance [27], we did not include the former among our covariates in the MLRM structures

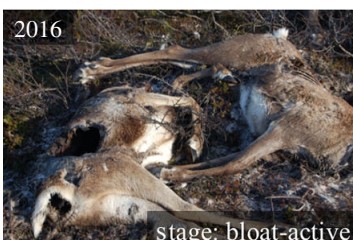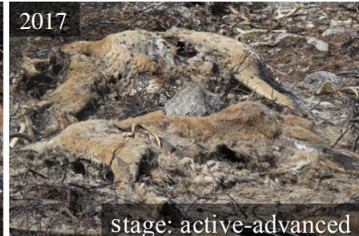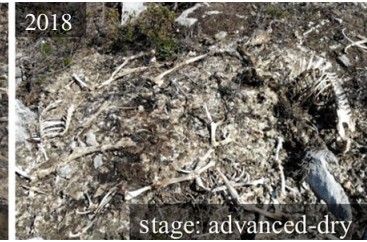

stage: bloat-active    stage: active-advanced    stage: advanced-dry

**Figure 2.** Carcass decomposition staging is depicted annually from 2016, when the mass die-off occurred on 26 August, until 2018. All pictures are from the autumn season and are staged according to a typical classification scheme provided by Barton & Bump [35]: 'fresh'—lasts only a few minutes; 'bloat'—describes the release of bacteria into gut, lymphatic and other tissues of the body along with the by-product of gases; 'active' decay—the putrefaction and liquefaction of carcass tissues, and the release of volatile organic compounds; 'advanced' decay—the final breakdown of soft tissues and the appearance of the skeleton; and 'dry' decay—occurs over a longer period, and can vary depending on the environment, but it refers to persistence of the ligaments, nails, hair and skeleton, whereas other carcass parts are absent. There was variability in the staging of carcasses on the study site, and most varied between two stages within a year (see 'stage' in images).

for modelling changes in rodent presence. Carcass biomass changed over the course of this study and we used a typical classification scheme [35] to identify their stage(s) as 'active to advanced' and 'advanced to dry' in 2017 and 2018, respectively (figure 2): 'active' refers to the decay and liquefaction of soft tissues; 'advanced' is the final breakdown of soft tissues and the appearance of the skeleton; and 'dry' refers to persistence of the ligaments, nails, hair and skeleton for a long period of time. As a result, available carrion biomass likely decreased, may have altered in distribution and probably influenced changes in scavenger use of the study site, but we did not have a direct estimate of this over time. However, we captured this at least partially by including (i) the carcass cover (of the plot) as a covariate (electronic supplementary material, figure S1) and (ii) indirectly through any changes in vegetation cover or study site use by mesopredators and birds. As rodents might also change their use of the study site due to interannual changes in vegetation or other cover types, we included covariates for the proportional cover of herbs, graminoids, lichen, moss, soil and stones (electronic supplementary material, figure S1). All model variables are continuous except for 'year' which was treated as a categorical factor. For changes in rodent presence, we did not have ecological justification for removing a variable from an MLRM global model *a priori*; we used the 'dredge' function from the MuMIn package in R [36] to fit a global model and Akaike's information criterion corrected for small sample sizes (AICc) to select the most parsimonious model for all candidate model sets (GLMs and MLRMs). A subset of model structures is shown in electronic supplementary material, table S1. The most complex MLRM global model that we considered for explaining changes in rodent presence was an additive model including mesopredator and avian abundance, and all cover types. The model with the lowest AICc value was considered the most parsimonious model. The observed spatial autocorrelation in response variables (e.g. up to 21 m for rodent presence, electronic supplementary material, figure S2) is likely due to the spatial autocorrelation (SAC) observed in our predictors (figure 1; electronic supplementary material, figure S1). To ensure that there was no bias in our inference, i.e. due to additional SAC beyond that observed in the predictors, we evaluated the influence of SAC on our results by using a Moran's *I* test on the residuals extracted for each response level of the most parsimonious models. If the most parsimonious model residuals depicted a significant Moran's *I* test *p*-value, we introduced a residual autocovariate term into model structures and re-fit the model. The residual autocovariate term accounts for the spatial autocorrelation of model residuals, e.g. based on neighbour networks [37,38]. We used the minimum distance among 'neighbours' possible which still achieved 'complete neighbour sets' to calculate neighbour networks. All final models were validated by visually inspecting residuals against covariates and fitted values, and fit was evaluated with goodness-of-fit tests and McFadden's pseudo-$R^2$. A pseudo-$R^2$ range between 0.2 and 0.4 indicates very good model fit [39]. We used R 3.5.2 for all statistical analyses and used an $\alpha$ level of 0.05 for testing significance [40].

## 3. Results

In total, we registered 44 mesopredator faeces ($N = 20$ in 2017, $N = 24$ in 2018) in 19 total unique plots ($N = 14$ in 2017, $N = 13$ in 2018), and 88 bird faeces ($N = 84$ in 2017, $N = 4$ in 2018) in 23 total unique plots ($N = 22$ in 2017, $N = 2$ in 2018). Given that faeces were detected, the mean count of faeces per plot was 1.6 for

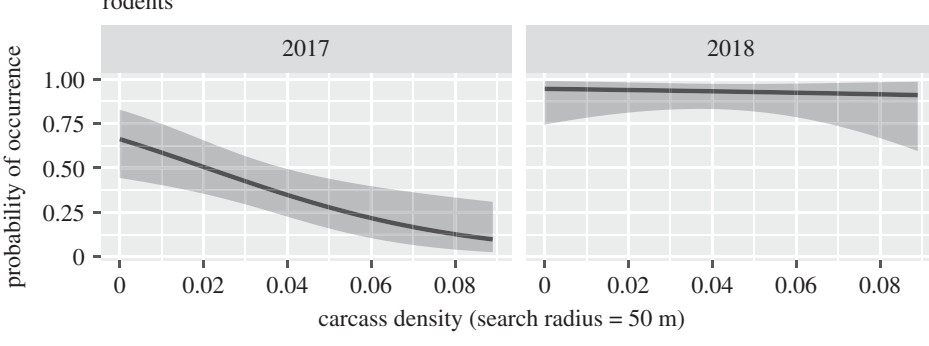

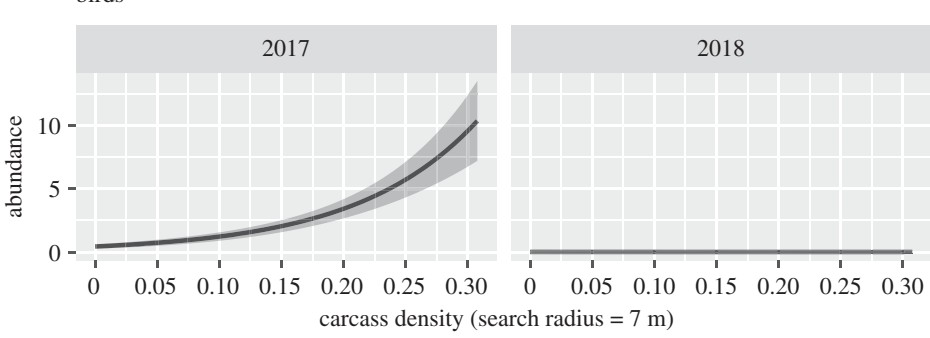

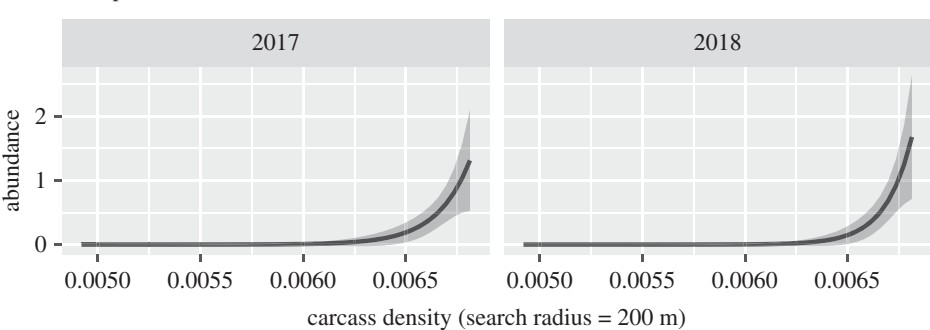

**Figure 3.** The analysis from Steyaert *et al*. [27] was adapted, i.e. only carcass density was used from a range of search radii as a covariate, and an interaction with year (2017 and 2018) was added to assess how each rodent presence, bird abundance and mammal abundance responses varied between years and with carcass density. Model selection (AICc) was used across each response and with the search radius of carcass density ranging (1, 2, 3, 4, 5, 6, 7, 8, 9, 10, 15, 20, 30, 40, 50, 100, 200 m). A residual autocovariate (RAC) was used in model structures for mesopredator and bird abundance responses, as those models depicted spatial autocorrelation (SAC; Moran's *I* test) in the residuals. Model selection was repeated and the most parsimonious models with RACs did not show SAC in the residuals (Moran's *I* test; *p*-value > 0.05). Rodent presence was significantly and negatively related with carcass density ($r = 50$ m; $\beta = -32.6$, s.e. = 11.5, *p*-value = 0.005) in 2017, but was independent of carcass density ($\beta = 26.5$, s.e. = 21.3, *p*-value = 0.212) and had a higher probability of occurrence in 2018 (ref year = 2017; $\beta = 2.19$, s.e. = 1.02, *p*-value = 0.033). Bird abundance was the opposite, with significantly higher abundance in higher carcass densities ($r = 7$ m; $\beta = 10.3$, s.e. = 0.904, *p*-value $\leq$ 0.001) and in 2017 (ref year = 2017; $\beta = -3.22$, s.e. = 1.45, *p*-value = 0.026) when compared with 2018 (also with no significant relationship with carcass density this year; $\beta = -16.8$, s.e. = 19.1, *p*-value = 0.379). For both years, mesopredator abundance was positively related to carcass density ($r = 200$ m; $\beta = -42.03$, s.e. = 11.96, *p*-value < 0.001) and did not significantly change across years (ref year = 2017, $\beta = 1563$, s.e. = 2652, *p*-value = 0.556). Model predictions across the observed ranges of carcass densities are shown above while holding the autocovariates at their mean values (for bird and mesopredator responses only).

mesopredators and 3.7 for birds. We detected faecal pellet groups of rodents at 56 unique plots ($N = 23$ in 2017, $N = 55$ in 2018). For mesopredators, use primarily occurred in the highest density of the carcasses (carcass density, 200 m: $\beta = 6210$, s.e. = 1786, *p*-value < 0.001; figure 3 and table 1) and appeared stable, with no significant change across years (year: $\beta = 10.18$, s.e. = 17.80, *p*-value = 0.567; figure 3 and table 1). For birds, use primarily occurred in the highest density of the carcasses in 2017 (carcass density, 7 m:

**Table 1.** Model coefficients estimates ($\beta$), their standard errors (s.e.), confidence intervals (CI) and significance ($p$-value) of the most parsimonious model for each response variable (rodent presence, bird and mesopredator abundance, and changes in rodent presence–absence). Significant covariates are bolded. All model variables are continuous except for 'year' which was treated as a categorical factor.

| response variable | distribution | coefficient name | $\beta$ | s.e. | CI | $p$-value |
|---|---|---|---|---|---|---|
| rodent presence | Bernoulli | intercept | 0.676 | 0.46 | [−0.209 to 1.62] | 0.142 |
| | | **carcass density ($r$ = 50 m)** | **−32.6** | **11.50** | **[−57.2 to −11.6]** | **0.005** |
| | | **year (2018)** | **2.19** | **1.02** | **[0.383 to 4.56]** | **0.033** |
| | | carcass density ($r$ = 50 m) × year (2018) | 26.5 | 21.30 | [−15.7 to 70.1] | 0.212 |
| bird abundance | Poisson | **intercept** | **−0.663** | **0.198** | **[−1.08 to −0.296]** | **<0.001** |
| | | **carcass density ($r$ = 7 m)** | **10.3** | **0.904** | **[8.56 to 12.1]** | **<0.001** |
| | | **RAC** | **0.205** | **0.0477** | **[0.112 to 0.299]** | **<0.001** |
| | | **year (2018)** | **−3.22** | **1.45** | **[−8.09 to −1.33]** | **0.026** |
| | | carcass density ($r$ = 7 m) × year (2018) | −16.8 | 19.1 | [n.a. to −0.334] | 0.379 |
| | | **RAC × year (2018)** | **1.43** | **0.672** | **[0.48 to 3.58]** | **0.034** |
| mesopredator abundance | Poisson | **intercept** | **−42.03** | **11.96** | **[−69.03 to −21.78]** | **<0.001** |
| | | **carcass density ($r$ = 200 m)** | **6210** | **1786** | **[3168.64 to 10 227.44]** | **<0.001** |
| | | RAC | 0.003 | 0.14 | [−0.28 to 0.27] | 0.983 |
| | | year (2018) | −10.18 | 17.80 | [−46.29 to 25.07] | 0.567 |
| | | carcass density ($r$ = 200 m) × year (2018) | 1563 | 2652 | [−3688.2 to 6937.74] | 0.556 |
| | | RAC × year (2018) | 0.295 | 0.18 | [−0.04 to 0.65] | 0.094 |
| changes in rodent presence–absence | multinomial (multiple response levels) | | | | | |
| | no change (reference level) | n.a. | n.a. | n.a. | n.a. | n.a. |
| | positive change (absence to presence) | intercept | −0.11 | 0.31 | [−0.71 to 0.489] | 0.718 |
| | **positive change (absence to presence)** | **bird** | **−0.469** | **0.23** | **[−0.922 to −0.0151]** | **0.0429** |
| | negative change (presence to absence) | intercept | −108 | 197.00 | [−494 to 278] | 0.582 |
| | negative change (presence to absence) | bird | −10.8 | 18.80 | [−47.6 to 26] | 0.566 |

$\beta = 10.3$, s.e. = 0.904, $p$-value < 0.001; electronic supplementary material, figure S3; table 1), but significantly dropped and was virtually absent in 2018 (year: $\beta = 3.22$, s.e. = 1.45, $p$-value = 0.026; electronic supplementary material, figure S3; table 1). By contrast, rodent use increased significantly between years (year: $\beta = 2.19$, s.e. = 1.02, $p$-value = 0.033; electronic supplementary material, figure S3, table 1) from virtually absent in the highest density of carcasses in 2017 to spanning nearly the entire study area in 2018 including high carcass density (carcass density, 50 m: $\beta = -32.6$, s.e. = 11.5, $p$-value = 0.005; electronic supplementary material, figure S3 and table S2). Of the 59 plots sampled, 58 had no or positive change in rodent presence, and only one plot depicted a change from rodent presence to absence. The most parsimonious model explaining the changes in rodent presence–absence included only changes in bird use as a predictor. Only positive changes in rodent presence–absence was significantly affected by changes in bird abundance (electronic supplementary material, figure S3; table 1; $\beta = -0.469$, s.e. = 0.231, $p$-value = 0.0429). Negative changes in rodent presence–absence were not significantly affected by changes in bird abundance (electronic supplementary material, figure S3; table 1; $\beta = -10.8$, s.e. = 18.8, $p$-value = 0.566). We detected significant spatial autocorrelation in the model residuals for the mesopredator abundance in 2018 ($p$-value < 0.001) and bird abundance response in both years (2017: $p$-value = 0.01, 2018: $p$-value < 0.001; electronic supplementary material, table S2), using a network with a minimum of 12 m distance among neighbours. After the inclusion of a residual autocovariate term in each candidate set, re-fitting and performing model selection, no evidence of SAC was detected (electronic supplementary material, table S2). All models had a 'good fit' as pseudo-$R^2$ values were all greater than 0.2 (electronic supplementary material, table S2). Goodness-of-fit tests were all insignificant except for changes in rodent presence–absence (electronic supplementary material, table S2), which was due to only one observation in the 'negative change' response level (electronic supplementary material, figure S3). If removed and fitted with a Bernoulli distribution, GOF tests are insignificant, and interpretation is virtually the same (aside from standard errors around minor changes to estimates; results not shown here). We chose to keep the full dataset and present those results for a fuller picture of the response. 'Changes in bird abundance' was the only covariate found to be important in predicting changes in rodent presence–absence across years, with a decrease in bird abundance linked to an increase in a positive change in rodent presence–absence (table 1; electronic supplementary material, figure S3).

## 4. Discussion

We found that (H1) scavenging birds, but not mesopredators, decreased their use of the study site from the first to the second year following an MME. Rodents, however, increased their use of the study site during this period and the high carcass density areas in particular. The increase of rodent use corresponded well with the virtual disappearance of birds during the second year, (H2) revealing evidence for a bird-induced LOF for rodents at reindeer carcasses. We suggest that rodents were released from fear of predation by the absence of corvids, not (mammalian) mesopredators.

Why did mesopredators not elicit the same apparent fear in rodents as did birds? The release from fear primarily occurred within the high carcass density areas, in which both mesopredators and birds occurred following the MME (our results and those in 27). Despite having another year of data, the final search radii for carcass density for each response were nearly identical to those from the study by Steyaert et al. [27] (rodent = 50 m for both, birds = 7 m for both, but mesopredators = 200 m here and 50 m in that study). This difference is likely due to the consistent intensive use of mesopredators in the high carcass density area. Given this consistent use by mesopredators, it is possible that the perceived predation risk by rodents was an additive effect from both mesopredator and bird use. For example, dogs and cats have been shown to jointly influence their prey's perception of foraging costs, but each individually failed to do the same [41]. LOFs are, however, typically driven by the diel activity of predators [30,42]. For example, even though raven and fox species both consume carrion and prey on rodents [34,43,44], foxes divide their active periods between day and night, with foxes on our study site appearing mostly nocturnal (camera trap data, Frank S, Steyaert SMJG, Nieland K, unpublished). Raven, however, are almost exclusively diurnal and appeared in the dozens at the study site ([45]; camera trap data, Frank S, Steyaert SMJG, Nieland K, unpublished; electronic supplementary material, figure S4). Moreover, most rodent species within the study area are diurnal (M. oeconomus [46], M. agrestis [47]) or crespuscular (M. glareolus; [48]) around our autumn sampling period, mirroring the active period for corvids at our study site (camera trap data, Frank S, Steyaert SMJG, Nieland K, unpublished). Lemmings deviate from this pattern and can be active both day and

night during this season [49]. Further, many more ravens than mesopredators visited the carcasses (camera trap data, unpublished, Frank S, Steyaert SMJG, Nieland K; electronic supplementary material, figure S4). Such droves of ravens presumably create a higher perceived mortality risk for rodents than the single or even occasionally several mesopredators (*sensu* [50]). Alternatively, if mortality risk is high in an attractive area and individuals do not perceive and respond to this risk, then an ecological trap could emerge, where individuals occur, but either die, e.g. from predation, or have reduced reproduction [51]. As more detailed information on space-use was not available for rodents and their predators, we were not able to distinguish between an LOF and ecological trap. Nevertheless, an ecological trap does not seem exclusively a cause or as likely as a LOF for our observed patterns for three reasons: (i) rodents are quite adept at perceiving risk and responding to it (e.g. [52–54]), which adds to the difficulty in managing them as pest populations [55,56], (ii) rodents were present around the periphery of the high carcass density in 2017 (figure 1), and (iii) ecological traps rarely result in 100% mortality [51]. The sheer amount of ravens in 2017 is hardly a risk that can either be passed as undetectable or ignored (electronic supplementary material, figure S4). A forensic study depicting scavenger taphonomic effects reported that in all cases rodents did not approach the carcass until other scavengers had left [57]. Home range sizes of the rodent species found within our study area ranged from 200 to greater than 5000 m$^2$ [58,59], making it likely that peripheral home ranges overlapped with the high carcass density core of the study site and indicating a behavioural response of rodents to actively avoid the high density carcass areas while raven were present in 2017. Although we were not able to disentangle an LOF and an ecological trap given our data, a landscape of fear and an ecological trap are not mutually exclusive; individual variation in rodent behavioural responses to the weighing of perceived risks against the benefits of an attractive resource could yield multiple survival outcomes. In essence, both could be at play in our study.

Additionally, rodent densities might have differed considerably between the 2 years of our study due to cycling populations [60,61], which we did not record. Such differences could have led to the observed differences in rodent presence between years, although the strong spatial pattern (avoidance of high carcass density areas in 2017) cannot be solely explained by altered rodent densities. Nevertheless, it is possible that rodents altered their space use of the study site due to changes in population density and other direct and indirect effects from carcasses, such as the interannual change in cover and therefore predation risk (e.g. decreasing carrion biomass and the return of vegetation may increase cover), and/or changes in food availability (e.g. the exposure of skeletal material and the return of vegetation) [31,53]. Further, the deposition of carcasses might create a suboptimal habitat that 'improves' over time for rodents, independent of predation risk, but soft tissue, rumen and skeletal material, which rodents can use (e.g. [31,32]), were available during both years and more so in 2017 when rodents occurred less. It is likely that vegetation cover would be responsive to changes in carcass biomass, i.e. some functional groups would persist and emerge (e.g. [62]), whereas others die out [2], and these vegetational shifts could provide a attractive habitat for rodents. Vegetation cover, however, did not appear to be important for changes in the presence–absence of rodents according to our models. Given that rodents are adept at perceiving risk and adjusting to risk, and that vegetation did not appear to be a good predictor for rodent presence–absence, it appears that other features correlated with carcass density may be drivers in the sudden appearance of rodents, i.e. the disappearance of birds. Indeed, carcass density is significantly associated with both rodent presence and bird abundance, and 'changes in bird abundance' was the best predicto for changes in rodent presence–absence. Alternatively, rodents may avoid fresh carcasses during the first year following the MME to avoid parasites [63,64], and return to the area the year after. Nevertheless, the risk of parasite infection may persist through the second year of carcass decomposition [65], and may not be an important factor within the timeframe of our study. Although disentangling potential confounding effects are virtually impossible with MMEs [7], studying them can yield insight into previously unreported processes.

Increasing carrion biomass can have nonlinear effects on (increasing) foraging efforts by insectivores [7]. In general, it is unclear how individual scavengers alter their foraging strategies in relation to carrion biomass along the continuum from single or few carcasses to mass mortalities, in addition to carcass size, and how these factors may modulate the presence of prey and predation risk for prey [7,28]. Understanding the carcass regime available to species, which could be described as the count, size (biomass), spatial distribution and deposition rate of carcasses [66], is crucial to understanding possible long-term, species-specific behaviour and fitness effects. Although predation risk can shape rodent behaviour and evolution [67], top-down effects on rodent population dynamics on landscapes appear to be secondary to bottom-up effects [60,68–70]. Even so, we provide evidence that the attraction of larger scavengers to an MME can greatly alter local space-use of smaller and more

vulnerable species, i.e. rodents. The degree to which a carcass regime affects individual behaviours and fitness (whether predator, scavenger, or prey) is an untapped field of research, but could provide answers for the management and persistence of wildlife populations.

Integrating the landscape of fear as a tool for managing rodent pest populations has recently gained attention [56] and it calls into question how humans can modulate a landscape of fear for animals, e.g. with supplemental feeding, bait sites and carcasses, to achieve management objectives in general. A more solid understanding of how carrion availability interacts with perceived risk by wildlife is required. The amount and distribution of carrion likely differs between natural and human-caused mortalities. In fact, most deaths of ungulates in managed systems appear to be human-caused, i.e. from hunting (e.g. [71–73]). Carrion created through hunting is often the leftovers (e.g. gut piles) following the removal of the majority of carcass biomass, whereas natural deaths other than that from predation leave bodies intact for scavengers to consume and from which necrobiomes can develop [74]. This difference is relevant for the management of ungulates, carnivores (carcass providers), small mammals and ecosystems as a whole, as predators and scavengers can influence one another's foraging strategies and certainly that of prey, i.e. landscapes of fear and even range expansion. For example, wolves provide carrion to scavengers, possibly paving the way for the latter's range expansion [75,76]. It is relatively unknown how recreational hunting could be used to emulate the same pathway towards conservation of target species. Ultimately though, the provisioning of large pulses of carcass biomass could be key to the persistence of populations and species, for example, during climatic shifts (e.g. [77]), but this strategy to artificially supply carrion is not without controversy [19,78]. Still, supplemental feeding is a common practice for maintaining biodiversity [79]. But, carrion in natural systems involve complexity that has not been explored in great detail. For example, species interactions can have a feedback on how carcass biomass is distributed on the landscape [80] and be responsible for the structure of ecosystems [81]. Understanding carcass provisioning, resulting landscapes of fear, and their effect on ecosystems and populations is lacking.

This study adds to our understanding of MMEs, i.e. that such large, ephemeral pulses of carrion can affect animal behaviour, not only through attraction, but also repulsion given a scavenger-induced LOF. It is unclear whether such scavenger-induced LOFs can affect population dynamics of smaller species or whether unpredictable MMEs can enhance fitness of larger scavengers (but see, e.g. [26]). Mass mortalities appear, in some cases, to be increasing [22] and may not be as rare as previously thought, as four MMEs were reported in Hardangervidda and surrounding mountain ranges within the last several years alone, and with no systematic search effort. This underscores the need to understand the role of MMEs in influencing animal behaviour, fitness and in structuring ecosystems. Further studies can be aimed towards estimating the actual frequency of MMEs within regions [66,82,83] and to assessing fitness effects along a continuum from single carcasses to mass mortality on both scavengers and other non-scavenger populations, and respective landscapes of fear within their vicinity.

Ethics. Landowner association 'Sameiet lisetfjell' and Telemark County provided fieldwork permission. No capture or handling of animals were carried out in this study.

Data accessibility. Carcass location, scat counts, study site boundary and R code used in this study are deposited in the Dryad Digital Repository: https://doi.org/10.5061/dryad.t×95×69sc [84].

Authors' contributions. S.C.F. and S.M.J.G.S. conceived the study. All authors assisted with fieldwork and data collection. S.C.F. and S.M.J.G.S. interpreted the data and SCF drafted the article. All authors provided valuable comments to earlier drafts, approved the manuscript, and agreed to be accountable for its content.

Competing interests. The authors declare no competing interests.

Funding. The REINCAR project is primarily self-funded, with additional financial assistance from National Geographic Society grant no. NGS-58488R-19 and from the LEBE! lab at the University of South-Eastern Norway.

Acknowledgements. REINCAR is hosted at the LEBE! lab at the University of South-Eastern Norway. We thank Christian Robstad and Frode Bergan for aiding in acquisition of fieldwork equipment and materials. We give a nod to the Norwegian Environment Agency for field guidance and the many volunteers who have helped in a multitude of ways since 2016. We thank Blue Oyster Cult for their original work and then Bruce Dickinson aand Gene Frenkle for adding more cowbell. We also thank two anonymous referees for comments that helped improve the manuscript.

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
