## [Reviewer comments · Royal Society Open Science]

Review History

RSOS-191644.R0 (Original submission)

Review form: Reviewer 1

Is the manuscript scientifically sound in its present form?

No

Are the interpretations and conclusions justified by the results?

Yes

Is the language acceptable?

Yes

Do you have any ethical concerns with this paper?

No

Have you any concerns about statistical analyses in this paper?

Yes

Recommendation?

Major revision is needed (please make suggestions in comments)

Comments to the Author(s)

This paper presents really interesting research that capitalized on a natural mass mortality event (MME) to ask how the landscape of fear from scavengers using the carcasses may influence small mammal distribution and use of the site. The study design is robust and the findings are really interesting, and it provides a novel examination of MME effects on other trophic levels by viewing them through the landscape of fear. Thus, I believe it's an important contribution to the growing body of literature on mass mortality events.

There are some areas where I feel the paper could be improved. The paper would benefit from the inclusion of more information about the distribution and decomposition stage of the carcasses in the two years of the study. If this information has already been described and presented elsewhere, then they should at least present a summary of it here. As is, it's difficult for the reader to have sufficient context to interpret the role carcasses may be playing. For example, the reindeer were killed in 2016, but this study examined the change in animal response to carcasses in 2017 and 2018 (a full one and two years later). Was there still a sufficient carcass signal to drive a large response in 2017 that would then have declined by 2018? It's difficult to judge this objectively without any information about how much carcass material remained available. Is it possible that some of other factor that differed between 2017 and 2018 (e.g., weather, etc) drove these patterns, and perhaps only influenced certain animal guilds?

I also had some questions about their statistical analysis. I did not understand their reasoning for not examining the influence of carcass density directly, although they then appeared to make conclusions about it, and I didn't understand their reasoning for several of the other steps involved in their analyses. More detailed comments are below:

- Li 57: was that the cutoff for 2015? >10 individuals?
- Li 114: It's not clear to me why you did not include these variables (carcass density and distance to nearest carcass) in modeling changes in rodent presence, even though you found they were positively correlated with scavenger abundance. Were you only looking at changes over time, and because they didn't change, you didn't think they were important?
- Li 116: It would be helpful if you could provide some kind of description of what proportion of the carcasses remained 1 and 2 years after the mortality event. Were there only bones remaining, or was there also dried skin and hair? How did this change to the second year? Or maybe include a photo as a figure. Even if you can't quantify this directly, it would be helpful to give the reader a bit more context, especially since the major goal of your study is to quantify the difference in response to this resource across the two years.
- Li 127: Doesn't AIC already consider parsimony, so why wouldn't you just use the model with the lowest AIC score?
- Li 144: How can you say that mesopredator use primarily occurred in the highest density of carcasses when you didn't actually test that factor?
- Li 149: Again, how do you know the relationship between rodent use and carcass density when you didn't test that specifically?
- Li 152: I do not understand this justification for dropping this datapoint. It seems that it's inclusion may increase the error around your model estimates, which would be important to include. Can you provide a better or more detailed justification for this?
- Li 155: The lack of spatial autocorrelation was surprising to me – was there no spatial autocorrelation in the distribution of carcasses? Can you provide some more info about this?
- Li 189: Do you mean decreasing carcass biomass?
- Li 192: It seems like you didn't "control" for vegetation cover as much as you analyzed its effect and found none, correct?
- Li 194: It is not clear how the second half of this sentence is a response to the questions raised in the prior sentences, and it isn't very clear on its own. Break this into its own sentence and explain your line of reasoning more clearly.
- Li 216: This is an interesting paragraph, but it doesn't seem directly related to your main findings about landscapes of fear and indirect effects of MMEs on small mammals. Is there a way you could make this more relevant to the paper's main findings?
- Fig. 2: I'm not clear what the values on the x axis represent. In the legend, it says these are the covariate for interannual changes in bird use of the area – it seems like this could be more clearly

indicated in the axis title. I'm also not sure how they decided where to calculate the 3 Pr points – do these represent particular values of interest, or are they just selected to provide general examples?

Review form: Reviewer 2

Is the manuscript scientifically sound in its present form?

Yes

Are the interpretations and conclusions justified by the results?

No

Is the language acceptable?

Yes

Do you have any ethical concerns with this paper?

No

Have you any concerns about statistical analyses in this paper?

No

Recommendation?

Major revision is needed (please make suggestions in comments)

Comments to the Author(s)

Comments to the Authors

It is a work in which the authors see that of a pulse of carrion that appears naturally and is an extraordinary event. It seems to be a very curious and interesting work due to the fact that it is massive mortality caused by lightning. Few research papers have the opportunity to examine the effect of a type of carrion contribution to the nature of this type. However, the issue of avoidance of prey in the carrion environment has been worked by other works such as the work done in Bialowieza.

In general, the work is considered correct.

Conceptually, after reading the work, I have a doubt whether what the authors find is 'landscape of fear' or is it an ecological trap for rodents.

Title: I suggest changing the title

Abstract

Line 16-17 redo

Line 140 mesopredator feces say how many of each species

There are some terms that I don't think are entirely correct, such as mesopredator feces, hotspots of species interactions, the hotspot of fear

Line 21 birdlife replace with ravens and foxes. Give some result

Avoidance towards avifauna, but not mammalian mesopredators major say what is corvids and foxes

Keywords

Not repeat the same words as in the title

Introduction

Hypothesis and predictions are not well raised. The authors propose predictions such as hypothesis. I think it would be more correct for them to raise the objective and the predictions or the general hypothesis and the predictions established in that framework. As it is now it is not correct.

H1 explain why

H2 with stool censuses I don't know how they can test this prediction.

H3 will not be because the rodents depend on the rumen or interior of the carrion and you go after the carrion has been opened and we are facing a case of facilitation or management in the consumption of carrion and everything else is voided by the LOF issue. In addition, I do not know what times of the year, small rodents do not explain the cycle of the authors but for what is described in the literature there are times that some spend them under the snow. Their activity underground is masked by the fallen snow. At the same time with high vegetation, the feces of many individuals have to be difficult to locate or even they can degrade quickly. In general, there is a lack of quality details about fieldwork

Methods

A map showing the location helps a reader who does not know how to locate the study. Figure 1 is not available to the reader.

Was it an open or closed area, hillside or flat?

How were the observations carried out?

How was the poop count carried out? Did they retire after counting them?

Line 103-104: The sampling period is not clear, if it was visited daily for a year or if it was only sampled those two days (putting ourselves at the end) More detail of the work is required.

Line 104-105: what categories were established

It is necessary to clarify well the methodology field, for example, the dates of the procedure because is not clear how many days/months took the monitoring. Also, the authors should provide more information about data collection by means of observations and sampling of feces.

Did the authors remove the feces? Also, more explanations are required on the categories established to measure the plant cover visually. In addition to all of these comments, it is required to explain if each those variables are calculated for each plot/sampling point

Analyzes:

It will be required that the authors define which are the response variables and which independent variables, which variable is considered random, categorical, continuous ... etc.

I wonder why they have not evaluated the probability of predatory prey coincidence per point based on the distance to the carcass site (Cortes-Avizanda et al 2019 work that the authors also cite).

Results

Line 140 provides data for each species.

Line 145: Is it correct to cite Figure 1?

Line 156: the results of the non-detection of autocorrelation must be explained before the results of the glms. I suggest to the authors that they better organize the presentation of the results.

A graphic with the temporal use of the carcasses would help to understand the results showing the raw data or he projections of the model

Table S1: the results as shown are redundant. If presence is cited, the number of presences detected and the number of times sampled should be shown, it is understood that they will be zero

I also believe that the authors could mention and discuss the other models that also have lower AICc

How much does the year explain in the total variance?

Discussion

In general, I believe that it is not fully demonstrated that it is fear and requires reinforcement. For instance, other factors may be playing (some already indicated by the authors). There are causes associated with the behavior of the species, it may be that there has been great predation at the beginning and then the rodents are not detected because the predation has been large, or it may be that it is linked to cycles of the voles, All this should be discussed.

Decision letter (RSOS-191644.R0)

08-Jan-2020

Dear Dr Frank,

The editors assigned to your paper ("Fear the reaper: ungulate carcasses generate an ephemeral landscape of fear for rodents") have now received comments from reviewers. We would like you to revise your paper in accordance with the referee and Associate Editor suggestions which can be found below (not including confidential reports to the Editor). Please note this decision does not guarantee eventual acceptance.

Please submit a copy of your revised paper before 31-Jan-2020. Please note that the revision deadline will expire at 00.00am on this date. If we do not hear from you within this time then it will be assumed that the paper has been withdrawn. In exceptional circumstances, extensions may be possible if agreed with the Editorial Office in advance. We do not allow multiple rounds of revision so we urge you to make every effort to fully address all of the comments at this stage. If deemed necessary by the Editors, your manuscript will be sent back to one or more of the original reviewers for assessment. If the original reviewers are not available, we may invite new reviewers.

- Data accessibility

<http://datadryad.org/submit?journalID=RSOS&manu=RSOS-191644>

- **Competing interests**

- **Authors' contributions**

- **Acknowledgements**

- **Funding statement**

Kind regards,

Andrew Dunn

on behalf of Prof Kevin Padian (Subject Editor)

Associate Editor's comments:

A number of queries, concerns, and comments have been raised by the reviewers of your manuscript. We'd like you to carefully consider these and provide complete responses (both in a point-by-point response document and, ideally, a tracked-changes version of the revised paper to aid both the Editors and reviewers in subsequent review). As the reviewers' comments are substantial, they will be invited to re-review if you choose to submit a revision. Thanks in advance.

Subject Editor comments:

Thanks for this most interesting submission, and best wishes for revising your MS.

Comments to Author:

Reviewers' Comments to Author:

Reviewer: 1

Comments to the Author(s)

This paper presents really interesting research that capitalized on a natural mass mortality event (MME) to ask how the landscape of fear from scavengers using the carcasses may influence small mammal distribution and use of the site. The study design is robust and the findings are really interesting, and it provides a novel examination of MME effects on other trophic levels by viewing them through the landscape of fear. Thus, I believe it's an important contribution to the growing body of literature on mass mortality events.

There are some areas where I feel the paper could be improved. The paper would benefit from the inclusion of more information about the distribution and decomposition stage of the carcasses in the two years of the study. If this information has already been described and presented elsewhere, then they should at least present a summary of it here. As is, it's difficult for the reader to have sufficient context to interpret the role carcasses may be playing. For example, the reindeer were killed in 2016, but this study examined the change in animal response to carcasses in 2017 and 2018 (a full one and two years later). Was there still a sufficient carcass signal to drive a large response in 2017 that would then have declined by 2018? It's difficult to judge this objectively without any information about how much carcass material remained available. Is it possible that some of other factor that differed between 2017 and 2018 (e.g., weather, etc) drove these patterns, and perhaps only influenced certain animal guilds?

I also had some questions about their statistical analysis. I did not understand their reasoning for not examining the influence of carcass density directly, although they then appeared to make conclusions about it, and I didn't understand their reasoning for several of the other steps involved in their analyses. More detailed comments are below:

- Li 57: was that the cutoff in 2015? >10 individuals?
- Li 114: It's not clear to me why you did not include these variables (carcass density and distance to nearest carcass) in modeling changes in rodent presence, even though you found they were positively correlated with scavenger abundance. Were you only looking at changes over time, and because they didn't change, you didn't think they were important?
- Li 116: It would be helpful if you could provide some kind of description of what proportion of the carcasses remained 1 and 2 years after the mortality event. Were there only bones remaining, or was there also dried skin and hair? How did this change to the second year? Or maybe include a photo as a figure. Even if you can't quantify this directly, it would be helpful to give the reader a bit more context, especially since the major goal of your study is to quantify the difference in response to this resource across the two years.
- Li 127: Doesn't AIC already consider parsimony, so why wouldn't you just use the model with the lowest AIC score?
- Li 144: How can you say that mesopredator use primarily occurred in the highest density of carcasses when you didn't actually test that factor?
- Li 149: Again, how do you know the relationship between rodent use and carcass density when you didn't test that specifically?
- Li 152: I do not understand this justification for dropping this datapoint. It seems that it's inclusion may increase the error around your model estimates, which would be important to include. Can you provide a better or more detailed justification for this?
- Li 155: The lack of spatial autocorrelation was surprising to me – was there no spatial autocorrelation in the distribution of carcasses? Can you provide some more info about this?
- Li 189: Do you mean decreasing carcass biomass?
- Li 192: It seems like you didn't "control" for vegetation cover as much as you analyzed its effect and found none, correct?
- Li 194: It is not clear how the second half of this sentence is a response to the questions raised in the prior sentences, and it isn't very clear on its own. Break this into its own sentence and explain your line of reasoning more clearly.

- Li 216: This is an interesting paragraph, but it doesn't seem directly related to your main findings about landscapes of fear and indirect effects of MMEs on small mammals. Is there a way you could make this more relevant to the paper's main findings?
- Fig. 2: I'm not clear what the values on the x axis represent. In the legend, it says these are the covariate for interannual changes in bird use of the area – it seems like this could be more clearly indicated in the axis title. I'm also not sure how they decided where to calculate the 3 Pr points – do these represent particular values of interest, or are they just selected to provide general examples?

Reviewer: 2

Comments to the Author(s)

Comments to the Authors

It is a work in which the authors see that of a pulse of carrion that appears naturally and is an extraordinary event. It seems to be a very curious and interesting work due to the fact that it is massive mortality caused by lightning. Few research papers have the opportunity to examine the effect of a type of carrion contribution to the nature of this type. However, the issue of avoidance of prey in the carrion environment has been worked by other works such as the work done in Bialowieza.

In general, the work is considered correct.

Conceptually, after reading the work, I have a doubt whether what the authors find is 'landscape of fear' or is it an ecological trap for rodents.

Title: I suggest changing the title

Abstract

Line 16-17 redo

Line 140 mesopredator feces say how many of each species

There are some terms that I don't think are entirely correct, such as mesopredator feces, hotspots of species interactions, the hotspot of fear

Line 21 birdlife replace with ravens and foxes. Give some result

Avoidance towards avifauna, but not mammalian mesopredators major say what is corvids and foxes

Keywords

Not repeat the same words as in the title

Introduction

Hypothesis and predictions are not well raised. The authors propose predictions such as hypothesis. I think it would be more correct for them to raise the objective and the predictions or the general hypothesis and the predictions established in that framework. As it is now it is not correct.

H1 explain why

H2 with stool censuses I don't know how they can test this prediction.

H3 will not be because the rodents depend on the rumen or interior of the carrion and you go after the carrion has been opened and we are facing a case of facilitation or management in the consumption of carrion and everything else is voided by the LOF issue. In addition, I do not know what times of the year, small rodents do not explain the cycle of the authors but for what is described in the literature there are times that some spend them under the snow. Their activity underground is masked by the fallen snow. At the same time with high vegetation, the feces of many individuals have to be difficult to locate or even they can degrade quickly. In general, there is a lack of quality details about fieldwork

Methods

A map showing the location helps a reader who does not know how to locate the study. Figure 1 is not available to the reader.

Was it an open or closed area, hillside or flat?

How were the observations carried out?

How was the poop count carried out? Did they retire after counting them?

Line 103-104: The sampling period is not clear, if it was visited daily for a year or if it was only sampled those two days (putting ourselves at the end) More detail of the work is required.

Line 104-105: what categories were established

It is necessary to clarify well the methodology field, for example, the dates of the procedure because is not clear how many days/months took the monitoring. Also, the authors should provide more information about data collection by means of observations and sampling of feces. Did the authors remove the feces? Also, more explanations are required on the categories established to measure the plant cover visually. In addition to all of these comments, it is required to explain if each those variables are calculated for each plot/sampling point

Analyzes:

It will be required that the authors define which are the response variables and which independent variables, which variable is considered random, categorical, continuous ... etc.

I wonder why they have not evaluated the probability of predatory prey coincidence per point based on the distance to the carcass site (Cortes-Avizanda et al 2019 work that the authors also cite).

Results

Line 140 provides data for each species.

Line 145: Is it correct to cite Figure 1?

Line 156: the results of the non-detection of autocorrelation must be explained before the results of the glms. I suggest to the authors that they better organize the presentation of the results.

A graphic with the temporal use of the carcasses would help to understand the results showing the raw data or he projections of the model

Table S1: the results as shown are redundant. If presence is cited, the number of presences detected and the number of times sampled should be shown, it is understood that they will be zero

I also believe that the authors could mention and discuss the other models that also have lower AICc

How much does the year explain in the total variance?

Discussion

In general, I believe that it is not fully demonstrated that it is fear and requires reinforcement. For instance, other factors may be playing (some already indicated by the authors). There are causes associated with the behavior of the species, it may be that there has been great predation at the beginning and then the rodents are not detected because the predation has been large, or it may be that it is linked to cycles of the voles, All this should be discussed.

Author's Response to Decision Letter for (RSOS-191644.R0)

See Appendix A.

RSOS-191644.R1 (Revision)

Review form: Reviewer 1

Is the manuscript scientifically sound in its present form?

Yes

Are the interpretations and conclusions justified by the results?

Yes

Is the language acceptable?

Yes

Do you have any ethical concerns with this paper?

No

Have you any concerns about statistical analyses in this paper?

No

Recommendation?

Accept with minor revision (please list in comments)

Comments to the Author(s)

The authors have done a good job responding to the points I raised in my review. I think their new statistical analyses are more robust, and their explanations are much clearer. I also like the new figures they have included. I have a few minor comments for consideration.

- I found it very helpful to see the comparison in temperature and rainfall data between 2017 and 2018 that the authors provided in their response to reviewers, but I didn't see it in the main text of the manuscript. I think this would be helpful to include in the Study Site section
- I really like Fig. S3 and think this is a helpful addition to the paper. I find it clearer and easier to interpret than the Fig. 3 currently included in the paper. I recognize they represent different analytical tests, but they are meant to convey a similar message. It seems the result of AIC analysis is most typically conveyed using a table, so I found Fig. 3 a little confusing to understand. I would suggest putting Fig. 3 in the SI and putting Fig. S3 and Table S2 in the main ms, although this is just a suggestion, and I leave this to the discretion of the authors.
- The addition of the photos in Fig. 2 and Fig. S4 are really helpful, as is the use of the carcass classification scheme to describe their decomposition stage.
- Li 278: I would re-phrase this as, "Given that rodents are adept at perceiving and adjusting to risk, and that vegetation did not appear to be a good predictor... , it appears that other features correlated..."

Decision letter (RSOS-191644.R1)

Dear Dr Frank:

On behalf of the Editors, I am pleased to inform you that your Manuscript RSOS-191644.R1 entitled "Fear the reaper: ungulate carcasses generate an ephemeral landscape of fear for rodents" has been accepted for publication in Royal Society Open Science subject to minor revision in accordance with the referee suggestions. Please find the referees' comments at the end of this email.

The reviewers and Subject Editor have recommended publication, but also suggest some minor revisions to your manuscript. Therefore, I invite you to respond to the comments and revise your manuscript.

- Ethics statement

If your study uses humans or animals please include details of the ethical approval received, including the name of the committee that granted approval. For human studies please also detail

whether informed consent was obtained. For field studies on animals please include details of all permissions, licences and/or approvals granted to carry out the fieldwork.

- Data accessibility

If you wish to submit your supporting data or code to Dryad (<http://datadryad.org/>), or modify your current submission to dryad, please use the following link:
<http://datadryad.org/submit?journalID=RSOS&manu=RSOS-191644.R1>

- Competing interests

- Authors' contributions

- Acknowledgements

- Funding statement

Because the schedule for publication is very tight, it is a condition of publication that you submit the revised version of your manuscript before 02-May-2020. Please note that the revision deadline will expire at 00.00am on this date. If you do not think you will be able to meet this date please let me know immediately.

on behalf of Kevin Padian (Subject Editor)
openscience@royalsociety.org

Associate Editor Comments to Author:

Comments to the Author:

We apologise for the delay with the processing of your manuscript submission. We have experienced difficulty in finding referees for your revision and have therefore opted to make a

decision based on the one report to avoid delaying your manuscript further. Please ensure that you address the remaining comments raised by the referee upon submitting your revision.

Reviewer comments to Author:

Reviewer: 1

Comments to the Author(s)

The authors have done a good job responding to the points I raised in my review. I think their new statistical analyses are more robust, and their explanations are much clearer. I also like the new figures they have included. I have a few minor comments for consideration.

- I found it very helpful to see the comparison in temperature and rainfall data between 2017 and 2018 that the authors provided in their response to reviewers, but I didn't see it in the main text of the manuscript. I think this would be helpful to include in the Study Site section
- I really like Fig. S3 and think this is a helpful addition to the paper. I find it clearer and easier to interpret than the Fig. 3 currently included in the paper. I recognize they represent different analytical tests, but they are meant to convey a similar message. It seems the result of AIC analysis is most typically conveyed using a table, so I found Fig. 3 a little confusing to understand. I would suggest putting Fig. 3 in the SI and putting Fig. S3 and Table S2 in the main ms, although this is just a suggestion, and I leave this to the discretion of the authors.
- The addition of the photos in Fig. 2 and Fig. S4 are really helpful, as is the use of the carcass classification scheme to describe their decomposition stage.
- Li 278: I would re-phrase this as, "Given that rodents are adept at perceiving and adjusting to risk, and that vegetation did not appear to be a good predictor... , it appears that other features correlated..."

Author's Response to Decision Letter for (RSOS-191644.R1)

See Appendix B.

Decision letter (RSOS-191644.R2)

Dear Dr Frank,

It is a pleasure to accept your manuscript entitled "Fear the reaper: ungulate carcasses may generate an ephemeral landscape of fear for rodents" in its current form for publication in Royal Society Open Science. The comments of the reviewer(s) who reviewed your manuscript are included at the foot of this letter.

You can expect to receive a proof of your article in the near future. Please contact the editorial

office (openscience_proofs@royalsociety.org) and the production office (openscience@royalsociety.org) to let us know if you are likely to be away from e-mail contact -- if you are going to be away, please nominate a co-author (if available) to manage the proofing process, and ensure they are copied into your email to the journal.

on behalf of Prof Kevin Padian (Subject Editor)
openscience@royalsociety.org

Appendix A

Dear Editor,

We think that this review process has greatly improved the work, as we received thoughtful and constructive comments. We believe our manuscript is now even of more interest to the readers of Royal Society Open Science, as it strengthens the case for how a mass die-off of reindeer can shape the landscape of fear for species, how carcasses can potentially be used for conservation and management, and how the role of carrion biomass (e.g. mass mortality events) play a role.

Our work is all original research carried out by the authors and is primarily self-funded. All authors have read and approved the contents of the updated manuscript in its revised form for submission to the journal Royal Society Open Science. This research has not been published in any form elsewhere. Our responses to the referees' comments follow.

Best regards,

Shane C. Frank on behalf of the co-authors.

Our response to referee comments:

Reviewer 1:

Comments to the Author(s)

This paper presents really interesting research that capitalized on a natural mass mortality event (MME) to ask how the landscape of fear from scavengers using the carcasses may influence small mammal distribution and use of the site. The study design is robust and the findings are really interesting, and it provides a novel examination of MME effects on other trophic levels by viewing them through the landscape of fear. Thus, I believe it's an important contribution to the growing body of literature on mass mortality events.

There are some areas where I feel the paper could be improved. The paper would benefit from the inclusion of more information about the distribution and decomposition stage of the carcasses in the two years of the study. If this information has already been described and presented elsewhere, then they should at least present a summary of it here. As is, it's difficult for the reader to have sufficient context to interpret the role carcasses may be playing. For example, the reindeer were killed in 2016, but this study examined the change in animal response to carcasses in 2017 and 2018 (a full one and two years later). Was there still a sufficient carcass signal to drive a large response in 2017 that would then have declined by 2018? It's difficult to judge this objectively without any information about how much carcass material remained available. Is it possible that some of other factor that differed between 2017 and 2018 (e.g., weather, etc) drove these patterns, and perhaps only influenced certain animal guilds?

***Our response:** We have now included more information about the decomposition stage of the carcasses in the two years of the study (please see below and Figure 2). In terms of the

distribution, they remained quite intact over the years (see new supplemental figure, Figure S1), with a subtle change depicted in distribution that is likely due to scavenging behavior, i.e., the moving of carcasses and pieces, and/or the decomposition process. We discuss this more in the text (Methods: Ln 130-146) Furthermore, the signal was strong enough in 2017 to attract a substantial number raven and, to give better context here, we included an example camera trap image from each year in additional supplemental figure (Figure S4). As for weather, mean temperatures and precipitation for the months of July and August (leading up to our visits) were very similar (2017-2018—precipitation: 2.73 – 2.92 mm, sd: 4.95 – 7.58; temperature: 10 – 10 C, sd: 0.756 – 0.802): <https://www.yr.no/en/statistics/>.

Reviewer 1: I also had some questions about their statistical analysis. I did not understand their reasoning for not examining the influence of carcass density directly, although they then appeared to make conclusions about it, and I didn't understand their reasoning for several of the other steps involved in their analyses.

***Our response:** We explain this in greater detail below (Comment at “Reviewer 1: Li 114”), but carcass density did not change *per se* across years. We now examine the relationship between carcass density and the presence of rodent scat, as well as the abundance of birds and mesopredators (see Figure S3). The effect of carcass ‘density’ or changes in carcass attributes on changes in rodent presence-absence was captured indirectly via animal use and vegetation cover of plots (Ln 124-134).

Reviewer 1:

• Li 57: was that the cutoff in Fey 2015? >10 individuals?

***Our response:** yes, that was the cutoff used by Fey et al. (2015) in relation to mammals. We make that clearer with “e.g. ≥ 10 individuals for mammals in Fey et al. 2015” (Ln61).

Reviewer 1:

• Li 114: It's not clear to me why you did not include these variables (carcass density and distance to nearest carcass) in modeling changes in rodent presence, even though you found they were positively correlated with scavenger abundance. Were you only looking at changes over time, and because they didn't change, you didn't think they were important?

***Our response:** Yes, we were only looking at changes over time. To explain further, we found a clear correlation between carcass density and distance to carcass with mammalian and avian feces in 2017 (Steyaert et al. 2018). We adapted those analyses to show how carcass density relates to the presence and abundance of fauna. Carcass density and distance to carcass did not change between 2017 and 2018, but carcass biomass due to decomposition (and consumption) very likely did. However, we did not document this at the individual carcass level, but rather as a basic description for the study area. Please see Figure S2 and Ln 124-134 for the information we added to give the reader better context, and a more detailed discussion with regards to the potential this may have had on rodent presence (Ln 262-284). Therefore, carcass density by itself would not explain “change” *per se* in rodent presence, but perhaps indicate that the change occurred in higher density areas (from absence to presence). Nevertheless, whatever changes that are occurring on the study area over time (which is certainly spatially linked with carcass density), such as changes in vegetative cover or presence of other species, are likely the driving forces of changes in rodent presence.

Therefore, we used predictor variables that actually changed over time to explain changes observed in rodent presence.

Reviewer 1:

- Li 116: It would be helpful if you could provide some kind of description of what proportion of the carcasses remained 1 and 2 years after the mortality event. Were there only bones remaining, or was there also dried skin and hair? How did this change to the second year? Or maybe include a photo as a figure. Even if you can't quantify this directly, it would be helpful to give the reader a bit more context, especially since the major goal of your study is to quantify the difference in response to this resource across the two years.

***Our response:** In the carcass decomposition description and Figure (S2), we included an image from 2016 (when the event occurred), 2017, and 2018 to give the reader more context. We used a 'typical classification' scheme provided by Barton and Bump (2019) to give general staging of the carcasses among years in Ln 134-139.

Reviewer 1:

- Li 127: Doesn't AIC already consider parsimony, so why wouldn't you just use the model with the lowest AIC score?

***Our response:** AICc does consider parsimony, and we therefore only present the model with the lowest AICc. Ln 154-155 reflects this change in 'Methods' and in the 'Results' (Table S1 and Figures 3 and S3)

Reviewer 1:

- Li 144: How can you say that mesopredator use primarily occurred in the highest density of carcasses when you didn't actually test that factor?

***Our response:** We deduced this from the findings in Steyaert et al. (2018) who showed that mesopredators occurred most at higher carcass densities in combination with our findings that there was no change in the distribution of mesopredator use across years (Fisher's exact test). We, however, have removed that analysis and now regressed the presence or abundance of species against carcass density and its interaction with year (Methods: Ln120-131), and have provided these results as supplemental information (Figure S3; Table S1-S3). The results are consistent with our previous conclusion that mesopredator abundance primarily occurred in the highest density of carcasses given that abundance has a significant, positive relationship with carcass density ($\beta = 6210$, P -value < 0.001) and that year and its interaction with density were not significant (P -value = 0.567 and 0.556, respectively).

Reviewer 1:

- Li 149: Again, how do you know the relationship between rodent use and carcass density when you didn't test that specifically?

***Our response:** Please refer also to the previous comment. There was a virtual absence of rodent presence in 2017 in the carcass dense area (Steyaert et al. 2018), and we detected a significant change in the distribution of rodent presence between years using a Fisher's Exact Test toward presence. We, however, added analyses (Figure 3, Tables S1-S3) and removed the contingency tables of counts (Fisher's Exact Test) and added a reference to Figure 3 and Tables S2 in the text (Ln 181-185), which clearly show that rodent presence changes from absence in the carcass dense area to a saturated presence in the carcass dense area (shaded

black). Moreover, year had a positive effect on rodent presence ($\beta = 2.19$, P -value = 0.033), indicating that presence increased from 2017-2018 (Table S2).

Reviewer 1:

- Li 152: I do not understand this justification for dropping this datapoint. It seems that it's inclusion may increase the error around your model estimates, which would be important to include. Can you provide a better or more detailed justification for this?

***Our response:** We initially used a multinomial model with 3 response levels for rodent presence ('no change': 0->0 | 0-> 1, 'positive change': 0 ->1, and 'negative change': 1->0), but there were convergence warnings, with resulting unreliable standard errors for the "negative change", and therefore unreliable for the "no change" and "positive change". We concluded that this was due to having only one observation in a response level. The model had difficulty in estimating uncertainty around this singular point in a response level, as there is no variation around it. We, however, revisited this analysis and increased the max iterations allowed by the multinom() function in R and the model converged. The reviewer is correct that standard errors changed due to the inclusion of this one point; they were larger for the 'positive change' and 'no change' responses, but the interpretation is nevertheless the same. We chose to show the multinomial model results alone, with the caveat that we have only one observation for one of the response levels, and explain that this causes a violation of the GOF test. From the analysis in the initial submission, we show that interpretation does not change in terms of direction and the GOF test yields a non-significant p-value when treated as a binary response, i.e. dropping the one observation for a negative change. We have amended the text to reflect these analyses in the results (Ln 197-202).

Reviewer 1:

- Li 155: The lack of spatial autocorrelation was surprising to me—was there no spatial autocorrelation in the distribution of carcasses? Can you provide some more info about this?

*** Our response:** There is certainly autocorrelation in the distribution of the carcasses. However, we aimed at reducing the effect of spatial autocorrelation from our gridded plot design and the response variable data gathered in relation to predictor variable data. As provided in the supplemental material (Figure S2), there is also spatial autocorrelation between plot location(s) and the response variables, which is partially how we determined a distance threshold in constructing 'neighborhoods' for assessing whether this spatial autocorrelation (SAC) depicted a pattern with the residuals of the 'best model'. If there was an indication of a pattern in the residuals, then we could assume that the SAC is problematic and introduced a residual autocovariate (RAC) into model structure, re-fit the model, and validated the model again using both Moran's I test, GOF, and McFadden's pseudo R^2 . Grant it, there is no true independence here, given that information gathered from the plots are to some degree related to one another, but we attempted to reduce the effect of SAC that would bias our interpretation by checking for its influence and setting about 'correcting' it as necessary. The results of these diagnostics (i.e. Moran's I test) following the introduction of RAC did not reveal further concern, i.e. a non-significant p-value, of the model residuals following correction (Table S3).

Reviewer 1:

- Li 189: Do you mean decreasing carcass biomass?

* **Our response:** Correct. To improve clarity, this is now changed to “decreasing carrion biomass” at Ln 274.

Reviewer 1:

• Li 192: It seems like you didn’t “control” for vegetation cover as much as you analyzed its effect and found none, correct?

* **Our response:** We corrected the text; it now reads “Vegetation cover, however, did not appear to be important...” at Ln276-278.

Reviewer 1:

• Li 194: It is not clear how the second half of this sentence is a response to the questions raised in the prior sentences, and it isn’t very clear on its own. Break this into its own sentence and explain your line of reasoning more clearly.

* **Our response:** We added clarifying text of the idea in Ln278-283. The updated analyses now help to clarify the relationship between density, rodent presence, and to each mesopredator and bird abundances. Those lines are shown below for convenience:

“Given that rodents are adept at perceiving risk, adjusting to risk, and vegetation did not appear to be a good predictor for rodent presence-absence, it appears that other features correlated with carcass density may be drivers in the sudden appearance of rodents, i.e., the disappearance of birds. Indeed, carcass density is significantly associated with both rodent presence and bird abundance, and changes in bird abundance was the best predictor for changes in rodent presence-absence.”

Reviewer 1:

• Li 216: This is an interesting paragraph, but it doesn’t seem directly related to your main findings about landscapes of fear and indirect effects of MMEs on small mammals. Is there a way you could make this more relevant to the paper’s main findings?

* **Our response:** We now better articulate the rationale for this paragraph (Ln 305-331). We establish that landscapes of fear are now becoming a tool for managing rodent populations and must be understood in relation to carrion availability. The paragraph is put here verbatim for convenience:

“Integrating the landscape of fear as a tool for managing rodent pest populations has recently gained attention (Krijger et al. 2017) and it calls into question how humans can modulate a landscape of fear for animals, e.g. with supplemental feeding, bait sites, and carcasses, to achieve management objectives in general. A more solid understanding of how carrion availability interacts with perceived risk by wildlife is required. Whether intentional or not, the amount and distribution of carrion likely differs between natural and human-caused mortalities. In fact, most deaths of ungulates in managed systems appear to be human-caused, i.e. from hunting (e.g. Bender et al. 2004, Brodie et al. 2013, Slabach et al. 2018). Carrion created through hunting is often the leftovers (e.g. gut piles) following the removal of the majority of carcass biomass, whereas natural deaths other than that from predation leave bodies intact for scavengers to consume and from which necrobiomes can develop (Pechal et al. 2013). This difference is relevant for the management of ungulates, carnivores (carcass providers), small mammals, and ecosystems as a whole, as predators and scavengers can influence one another’s foraging strategies and certainly that of prey, i.e. landscapes of fear and even range expansion. For example, wolves provide carrion to scavengers, possibly

paving the way for the latter's range expansion (Wilmers et al. 2003, van Dijk et al. 2008). It is relatively unknown how recreational hunting could be used to emulate the same pathway toward conservation of target species. Ultimately though, the provisioning of large pulses of carcass biomass could be key to the persistence of populations and species, for example, during climatic shifts (e.g. Laidre et al. 2018), but this strategy to artificially supply carrion is not without controversy (Cortés-Avizanda et al. 2009a, Cortés-Avizanda and Pereira 2016). Still, supplemental feeding is a common practice for maintaining biodiversity (Ewen et al. 2015). But, carrion in natural systems involve complexity that has not been explored in great detail. For example, species interactions can have a feedback on how carcass biomass is distributed on the landscape (Harding et al. 2019) and be responsible for the structure of ecosystems (Everatt et al. 2016). Understanding how carcass provisioning, resulting landscapes of fear, and their effect on ecosystems and populations is lacking.”

Reviewer 1:

- Fig. 2: I'm not clear what the values on the x axis represent. In the legend, it says these are the covariate for interannual changes in bird use of the area—it seems like this could be more clearly indicated in the axis title. I'm also not sure how they decided where to calculate the 3 Pr points—do these represent particular values of interest, or are they just selected to provide general examples?

*** Our response:** We changed the analysis to a multinomial model and also changed the Figure (now Figure 3) with the x-axis that now reads “Interannual changes in bird abundance (2017-2018)”.

Reviewer 2: It is a work in which the authors see that of a pulse of carrion that appears naturally and is an extraordinary event. It seems to be a very curious and interesting work due to the fact that it is massive mortality caused by lightning. Few research papers have the opportunity to examine the effect of a type of carrion contribution to the nature of this type. However, the issue of avoidance of prey in the carrion environment has been worked by other works such as the work done in Bialowieza.

***Our response:** To our knowledge only very few publications exist on this topic. In addition, we could not find evidence of a landscape of fear framework being applied to a mass die-off, both of which we mention in the MS (Ln 53-57 and 65-66).

In general, the work is considered correct.

Reviewer 2: Conceptually, after reading the work, I have a doubt whether what the authors find is ‘landscape of fear’ or is it an ecological trap for rodents.

***Our response:** This doubt is justified, however our results support more strongly that a landscape of fear is present compared to an ecological trap. Even though carcasses can be attractive for rodents, rodents are very adept at perceiving and avoiding risk, and they were present throughout the study (and with home range sizes that likely overlap the carcass area). We now articulate this rationale above and address in the manuscript that we cannot completely rule an ecological trap (Ln 239-260). The specific paragraph follows: “Alternatively, if mortality risk is high in an attractive area and individuals do not perceive and respond to this risk, then an ecological trap could emerge, where individuals occur, but either die, e.g., from predation, or have reduced reproduction (Hale and Swearer 2016). As

more detailed information on space-use was not available for rodents and their predators, we were not able to distinguish between a LOF and ecological trap. Nevertheless, an ecological trap does not seem exclusively a cause or as likely as a LOF for our observed patterns for three reasons: (1) rodents are quite adept at perceiving risk and responding to it (e.g. Orrock 2004, Suraci et al. 2019, Bleicher et al. 2020), which adds to the difficulty in managing them as pest populations (Krijger et al. 2017, Bedoya-Pérez et al. 2019), (2) rodents were present around the periphery of the high carcass density in 2017 (Figure 1), and (3) ecological traps rarely result in 100% mortality (Hale and Swearer 2016). The sheer amount of ravens in 2017 is hardly a risk that can either be passed as undetectable or ignored (Figure S4). A forensic study depicting scavenger taphonomic effects reported that in all cases rodents did not approach the carcass until other scavengers had left (Komar and Beattie 1998). Home range sizes of the rodent species found within our study area range from 200 to >5000 m² (Lambin et al. 1992, Fauske et al. 1997), making it likely that peripheral home ranges overlapped with the high carcass density core of the study site and indicating a behavioural response of rodents to actively avoid the high density carcass areas whilst raven were present in 2017. Although we were not able to disentangle a LOF and an ecological trap given our data, a landscape of fear and an ecological trap are not mutually exclusive; individual variation in rodent behavioural responses to the weighing of perceived risks against the benefits of an attractive resource could yield multiple survival outcomes. In essence, both could be at play in our study.”

Reviewer 2: Title: I suggest changing the title

***Our response:** we are not sure what the basis is for changing the title, but we assume the Reviewer is referring to whether this is a LOF or ecological trap. We ‘soften’ the title with ‘may’, still indicating that there is evidence for a LOF here, but also admitting that it could be an ecological trap (see previous answer or in the Discussion: Ln 239-260).

Abstract

Reviewer 2: Line 16-17 redo

***Our response:** We are not clear on what the Reviewer intends with “redo” for these few lines. We have reworded it, in hopes that it has improved and “hotspot” has been removed as the Reviewer mentioned it as being misused. It now reads “Carcass sites can increase species interactions and/or ephemeral, localized landscapes of fear for prey within the vicinity.” (Ln 17-18).

Reviewer 2: Line 140 mesopredator feces say how many of each species

***Our response:** we could not tell the difference between arctic fox and red fox by sight, nor did we take them for lab for genetic analysis, but their dietary niches are quite similar and with rodents as a common component of their diets (Elmhagen, B., et al. (2002). "Food-niche overlap between arctic and red foxes." *Canadian Journal of Zoology* 80(7): 1274-1285). We purposefully left scat in-tact for another on-going study.

Reviewer 2: There are some terms that I don't think are entirely correct, such as mesopredator feces, hotspots of species interactions, the hotspot of fear

***Our response:** We changed “hotspot” to indicate an area that can “increase species interactions” (Ln 17) and changed “hotspot for fear” to “localized landscapes of fear for prey.” (Ln 18). This is also changed in the main text (Ln 43-45).

Reviewer 2: Line 21 birdlife replace with ravens and foxes. Give some result Avoidance towards avifauna, but not mammalian mesopredators major say what is corvids and foxes

***Our response:** Changed taxa names and added the beta coefficient, SE, and p-value ($\beta = -0.469$, $SE = 0.231$, $p\text{-value} = 0.0429$) depicting that rodents avoided changes in raven abundance (Ln 20-23).

Reviewer 2: Keywords

Not repeat the same words as in the title

***Our response:** Removed “landscape of fear” and “rodents”, and added “carcass decomposition” and “fear ecology” (Ln 33-34)

Reviewer 2: Introduction

Hypothesis and predictions are not well raised. The authors propose predictions such as hypothesis. I think it would be more correct for them to raise the objective and the predictions or the general hypothesis and the predictions established in that framework. As it is now it is not correct.

***Our response:** We added more explicit hypotheses and general predictions (Ln 71-75) and go on to detail predictions specific to our data/analyses in support of the proposed hypotheses.

Reviewer 2: H1 explain why

***Our response:** We added “i.e. soft tissues that are more attractive to raven and fox” to better describe the depleted resources (Ln 77-78). In addition, the staging of carcasses (Figure 2) helps put this into better context in the Methods.

Reviewer 2: H2 with stool censuses I don't know how they can test this prediction.

***Our response:** We recognize the limitations of these data and adjusted the title and addressed the potential for an ecological trap in the Discussion (Ln 240-261). However, it is highly unlikely that (1) rodents did not perceive the risk of raven (please see Figure S4), (2) that raven would be so efficient as to not have one rodent leave behind a single fecal pellet group within the high carcass density area, and (3) that rodent home ranges from 2017 did not overlap with the carcass area. We agree, however, that mortality occurred, which is a key component to instilling ‘fear’ in prey, but do not concur that it is solely an ecological trap. It is more likely to be fear, reinforced by mortality, therefore a combination of both. Even though it is not possible to completely disentangle a LOF and an ecological trap, given our data, we contend that it is more likely to be a landscape of fear or a mixture of the two, rather than an ecological trap alone.

Reviewer 2: H3 will not be because the rodents depend on the rumen or interior of the carrion and you go after the carrion has been opened and we are facing a case of facilitation or management in the consumption of carrion and everything else is voided by the LOF issue.

***Our response:** We understand how the comment can arise without proper context on the carcass stages over the years. We think that the carcass staging Figure (Figure 2) will help dispel this concern. Substantial rumen (active-advanced) was available in 2017 and to a lesser degree (advanced-dry) in 2018 (Figure 2). Therefore, the facilitation would have been ‘stronger’ in 2017, not 2018, but the point is that there was available rumen during both years for rodents. We added text at Ln 134-141 indicating that the availability of food sources for each of the animal taxa decreased over time, but was still available:

“Carcass biomass changed over the course of this study and we used a typical classification scheme (Barton and Bump 2019) to identify their stage(s) as ‘active to advanced’ and ‘advanced to dry’ in 2017 and 2018, respectively (Figure 2): “active” refers to the decay and liquefaction of soft tissues; ‘advanced’ is the final breakdown of soft tissues and the appearance of the skeleton; and ‘dry’ refers to persistence of the ligaments, nails, hair, and skeleton for a long period of time. As a result, available carrion biomass likely decreased, may have altered in distribution, and probably influenced changes in scavenger use of the study site, but we did not have a direct estimate of this over time.”

Reviewer 2: In addition, I do not know what times of the year, small rodents do not explain the cycle of the authors but for what is described in the literature there are times that some spend them under the snow. Their activity underground is masked by the fallen snow. At the same time with high vegetation, the feces of many individuals have to be difficult to locate or even they can degrade quickly. In general, there is a lack of quality details about fieldwork

***Our response:** As pointed out in the methods (Ln 113-114), we only sampled feces one day per year at the study site, as dates (i.e. 11 August 2017 and 4 August 2018) were given for the fieldwork. There was no snow cover during this period and visibility was good for conducting our searches. Furthermore, we gave how much time was spent on each plot (4 minutes x 59 plots). Unfortunately, it was not feasible to do more sampling given the remoteness of the location. It is possible that feces could have degraded differentially between years rendering the cumulative sampling that we did different across years, but we do not believe this to be the case, as mean temperatures and precipitation for the months of July and August (leading up to our visits) were very similar (2017-2018—precipitation: 2.73 – 2.92 mm, sd: 4.95 – 7.58; temperature: 10 – 10 C, sd: 0.756 – 0.802):

<https://www.yr.no/en/statistics/>. Nevertheless, it is possible that rodent densities were different across years, due to rodent population cycles, but it would not likely explain the differences in spatial use. We discuss this in Ln 262-266:

“Additionally, rodent densities might have differed considerably between the two years of our study due to cycling populations (Selås 2016, Johnsen et al. 2019), which we did not record. Such differences could have led to the observed differences in rodent presence between years, although the strong spatial pattern (avoidance of high carcass density areas in 2017) cannot be solely explained by altered rodent densities.”

Reviewer 2: Methods

A map showing the location helps a reader who does not know how to locate the study.

***Our response:** A map was added to Figure 1

Reviewer 2: Figure 1 is not available to the reader.

***Our response:** We do not understand this comment, as Figure 1 is in the document. We added a map to the Figure 1 to clarify the location of the study site.

Reviewer 2: Was it an open or closed area, hillside or flat?

***Our response:** We added this information to lines 88-89 and reads "...and with a small north-northeast facing slope."

Reviewer 2: How were the observations carried out?

***Our response:** This information can be found on lines 108-118. It reads:

"We set up a 10 × 10 m grid containing 59 1 × 1 m survey plots covering the study site (Figure 1). Each plot was subdivided into four 50 × 50 cm quadrants in which two observers systematically searched each for 30 s, totalling 4 minutes per plot, and recorded the number of (i) mammalian mesopredator faeces (i.e. red and arctic fox, hereafter 'mesopredators'), (ii) bird faeces, and (iii) the presence of rodent faecal pellet groups (≥1 fecal pellet detected). All faeces were left *in situ* for other on-going studies. The same two observers collected these plot-level data for two consecutive years during autumn on single day visits: 11 August 2017 and 4 August 2018. In addition, detailed cryptogams that included plants identified to species or genus level for each year were conducted for another on-going study and species were clustered into functional groups including vascular plants, graminoids, mosses, and lichens. Percent coverage for these plants and for soil, stone, and carcass for the northwest quadrant of each plot were estimated visually."

Reviewer 2: How was the poop count carried out?

***Our response:** This information can be found at lines 109-114 and reads:

"Each plot was subdivided into four 50 × 50 cm quadrants in which two observers systematically searched each for 30 s, totalling 4 minutes per plot, and recorded the number of (i) mammalian mesopredator faeces (i.e. red and arctic fox, hereafter 'mesopredators'), (ii) bird faeces, and (iii) the presence of rodent faecal pellet groups (≥1 fecal pellet detected). All faeces were left *in situ* for other on-going studies. The same two observers collected these plot-level data for two consecutive years during autumn on single day visits: 11 August 2017 and 4 August 2018."

Reviewer 2: Did they retire after counting them?

***Our response:** Over the course of a year between counts, we assumed that scat disintegrated and disappeared, which is likely given the harsh environment (melting, freezing, etc.), but they were stable enough for our observations to be cumulative for an unknown period. We, however, do not know their persistence. Consistent temperature and moisture patterns within ~2 months up to our sampling dates (see comments above) indicate that scat persistence would be similar across years and given that our sampling days of the year were very close to one another.

Reviewer 2: Line 103-104: The sampling period is not clear, if it was visited daily for a year or if it was only sampled those two days (putting ourselves at the end) More detail of the work is required.

***Our response:** We have improved clarity by adding "single day visits" (Ln 114).

Reviewer 2: Line 104-105: what categories were established

It is necessary to clarify well the methodology field, for example, the dates of the procedure because is not clear how many days/months took the monitoring. Also, the authors should provide more information about data collection by means of observations and sampling of feces. Did the authors remove the feces? Also, more explanations are required on the categories established to measure the plant cover visually. In addition to all of these comments, it is required to explain if each those variables are calculated for each plot/sampling point

***Our response:** We clarified the time spent sampling and the ‘handling’ of feces and how plant cover was estimated above for the comment entitled “Reviewer 2: How were the observations carried out?”. This information can be found in the main text at lines 108-118. We added ‘plot-level’ at Ln 114, because essentially all variables were scaled to the plot level.

Reviewer 2: Analyzes:

It will be required that the authors define which are the response variables and which independent variables, which variable is considered random, categorical, continuous ... etc. I wonder why they have not evaluated the probability of predatory prey coincidence per point based on the distance to the carcass site (Cortes-Avizanda et al 2019 work that the authors also cite).

***Our response:** At Ln146-147 and in table captions (Tables S1-S2) we now clarify that all model variables are continuous except for ‘year’ which was treated as a categorical factor. With regards to modeling co-occurrence instead, we see this as giving little difference in outcome for interpretation. Our analytical strategy goes directly at whether changes in bird abundance are responsible for changes in rodent presence-absence, in addition to the other changes such as plant cover and mesopredator abundance. The assumptive inference from the oc-occurrence analytical strategy would likely be that predation ‘risk’ is high for rodents from mesopredators and low from birds. Understandably, this is from the standpoint of high mortality rather than high avoidance. Again, given the support from the literature, we think that a LOF is more likely than an ecological trap (high attractiveness met with high mortality), and so choose to keep our framework. Nevertheless, we recognize and appreciate the suggested method from that perspective.

Reviewer 2: Results

Line 140 provides data for each species.

***Our response:** At Ln147-148 and in table captions (Tables S1-S2) we now clarify that all model variables are continuous except for ‘year’ which was treated as a categorical factor.

Reviewer 2: Line 145: Is it correct to cite Figure 1?

***Our response:** Figure 1 is more descriptive of design, so we corrected this to reference Figure S3 and Table S2 in lines 178, 179, and 182.

Reviewer 2: Line 156: the results of the non-detection of autocorrelation must be explained before the results of the glms. I suggest to the authors that they better organize the presentation of the results.

***Our response:** Semivariograms provided in the Supplemental Information (Figure S2) show the autocorrelation with response variables, but GLMs must be fitted to evaluate whether autocorrelation of the plot design affected the results, i.e., depicting a pattern with model residuals. We now better explain how we used the semivariograms in the methods (Ln 155-165). This now reads:

“The observed spatial autocorrelation in response variables (e.g. up to 21 m for rodent presence, Figure S2) is likely due to the spatial autocorrelation (SAC) observed in our predictors (Figure 1 and Figure S1). To ensure that there was no bias in our inference, i.e. due to additional SAC beyond that observed in the predictors, we evaluated the influence of SAC on our results by using a Moran’s I test on the residuals extracted for each response level of the most parsimonious models. If the most parsimonious model residuals depicted a significant Moran’s I test *P*-value, we introduced a residual autocovariate term into model structures and re-fit the model. The residual autocovariate term accounts for the spatial autocorrelation of model residuals, e.g. based on neighbour networks (Dormann et al. 2007, Crase et al. 2012). We used the minimum distance among ‘neighbours’ possible which still achieved ‘complete neighbour sets’ to calculate neighbour networks.”

Reviewer 2: A graphic with the temporal use of the carcasses would help to understand the results showing the raw data or the projections of the model

***Our response:** We have added these analyses (Methods: Ln 120-169, Results: 176-186, Figure S3, Table S1-S3). As figures are limited in the main text, we included these results in the Supplements. We removed the Fisher Exact Tests. We think this helps our case and conclusion about animal use of carcass density.

Reviewer 2: Table S1: the results as shown are redundant. If presence is cited, the number of presences detected and the number of times sampled should be shown, it is understood that they will be zero

***Our response:** Table S1, i.e. the contingency table, in the supplemental from the previous version has been removed and the Fisher’s Exact Test is not used.

Reviewer 2: I also believe that the authors could mention and discuss the other models that also have lower AICc

***Our response:** We have updated our analyses and results, and the most recent results depicting changes in rodent presence-absence now do not include other models that fall within a delta AICc of 2 (new Table S1). We show models AICc ≤ 4 . The exception is the search radius for carcass density in relation to each of the responses, as several delta AICc values are within 2 due to their high correlation. Nevertheless, we compare the resulting carcass density search radii in the final models with those from Steyaert et al. (2018) in lines 218-222, and it reads:

“Despite having another year of data, the final search radii for carcass density for each response were nearly identical to those from the study by Steyaert et al. (2018) (rodent = 50 m for both, birds = 7 m for both, but mesopredators = 200 m here and 50 m in that study). This difference is likely due to the consistent intensive use of mesopredators in the high carcass density area.”

Reviewer 2: How much does the year explain in the total variance?

***Our response:** This is not pertinent for the ‘change analysis’, as year is implicit in the data set, but it now can be expressed in our new model structure/analyses on the effect of carcass density on presence and abundance responses. Nevertheless, we use McFadden’s Pseudo R^2 (Methods: Ln 167-168) as a measure of explained deviance for the most parsimonious models (Table S3) and do not isolate ‘year’ in a model that was not selected with this metric. The main text lines read as:

“All final models were validated by visually inspecting residuals against covariates and fitted values, fit was evaluated with goodness-of-fit tests and McFadden’s Pseudo R^2 . A pseudo R^2 range between 0.2 and 0.4 indicates very good model fit (Lee 2013).”

Reviewer 2: Discussion

In general, I believe that it is not fully demonstrated that it is fear and requires reinforcement. For instance, other factors may be playing (some already indicated by the authors). There are causes associated with the behavior of the species, it may be that there has been great predation at the beginning and then the rodents are not detected because the predation has been large, or it may be that it is linked to cycles of the voles All this should be discussed.

***Our response:** In general, we agree with this critique and soften our conclusions (both in title and in discussion). Nevertheless, we provide citations indicating that rodents are quite perceptive to risk and have several litters per year, suggesting that initial depressions in individuals could be met with recruiting individuals and/or presence. Furthermore, there was not a complete absence of rodents in the area in 2017 and example home range sizes of species range from 200 to $>5000 \text{ m}^2$, suggesting that these rodents’ home ranges likely overlapped with the high carcass density area. Therefore, although we agree that mortality likely increased for rodents, fear is also very likely to have played a pivotal role given our rationale above. Furthermore, rodent population cycling could explain a difference in density of rodents across years, but less likely to explain the spatial patterns we see on the carcass site, to which fear is a better explanation. These points have been raised in the Discussion (Ln 262-290) and reads:

“Additionally, rodent densities might have differed considerably between the two years of our study due to cycling populations (Selås 2016, Johnsen et al. 2019), which we did not record. Such differences could have led to the observed differences in rodent presence between years, although the strong spatial pattern (avoidance of high carcass density areas in 2017) cannot be solely explained by altered rodent densities. Nevertheless, it is possible that rodents altered their space use of the study site due to changes in population density and other direct and indirect effects from carcasses, such as the interannual change in cover and therefore predation risk (e.g. decreasing carrion biomass and the return of vegetation may increase cover), and/or changes in food availability (e.g. the exposure of skeletal material and the return of vegetation) (Orrock 2004, Young et al. 2014). Further, the deposition of carcasses might create suboptimal habitat that ‘improves’ over time for rodents, independent of predation risk, but soft tissue, rumen, and skeletal material, which rodents can utilize (e.g. Young et al. 2014, Pokines 2015), were available during both years and more so in 2017 when rodents occurred less. It is likely that vegetation cover would be responsive to changes in carcass biomass, i.e., some functional groups would persist and emerge (e.g. Barton et al. 2016), whereas others die out (Benbow et al. 2015), and these vegetational shifts could provide attractive habitat for rodents. Vegetation cover, however, did not appear to be important for changes in the presence-absence of rodents according to our models. Given that

rodents are adept at perceiving risk, adjusting to risk, and vegetation did not appear to be a good predictor for rodent presence-absence, it appears that other features correlated with carcass density may be drivers in the sudden appearance of rodents, i.e., the disappearance of birds. Indeed, carcass density is significantly associated with both rodent presence and bird abundance, and changes in bird abundance was the best predictor for changes in rodent presence-absence. Alternatively, rodents may avoid fresh carcasses during the first year following the MME to avoid parasites (Weinstein et al. 2018a, Weinstein et al. 2018b), and return to the area the year after. Nevertheless, the risk of parasite infection may persist through the second year of carcass decomposition (Turner et al. 2014), and may not be an important factor within the timeframe of our study. Although disentangling potential confounding effects are virtually impossible with MMEs (Lashley et al. 2018), studying them can yield insight to previously unreported processes.”

Appendix B

Dear Editor,

Thank you for the consideration and acceptance of this article.

Our responses to the referee's comments follow * below.

Best regards,

Shane C. Frank on behalf of the co-authors.

Associate Editor Comments to Author:

Comments to the Author:

We apologise for the delay with the processing of your manuscript submission. We have experienced difficulty in finding referees for your revision and have therefore opted to make a decision based on the one report to avoid delaying your manuscript further. Please ensure that you address the remaining comments raised by the referee upon submitting your revision.

***Our response:** we understand that difficulties arise with finding reviewers. Thank you for the apology. We do our best to contribute as reviewers, but admittedly cannot always do so ourselves.

Reviewer comments to Author:

Reviewer: 1

Comments to the Author(s)

The authors have done a good job responding to the points I raised in my review. I think their new statistical analyses are more robust, and their explanations are much clearer. I also like the new figures they have included. I have a few minor comments for consideration.

- I found it very helpful to see the comparison in temperature and rainfall data between 2017 and 2018 that the authors provided in their response to reviewers, but I didn't see it in the main text of the manuscript. I think this would be helpful to include in the Study Site section

***Our response:** The weather data across years and during focal months of July-August are now included in lines 118-121.

- I really like Fig. S3 and think this is a helpful addition to the paper. I find it clearer and easier to interpret than the Fig. 3 currently included in the paper. I recognize they represent different analytical tests, but they are meant to convey a similar message. It seems the result of AIC analysis is most typically conveyed using a table, so I found Fig. 3 a little confusing

to understand. I would suggest putting Fig. 3 in the SI and putting Fig. S3 and Table S2 in the main ms, although this is just a suggestion, and I leave this to the discretion of the authors.

***Our response:** we opted to do as the reviewer suggested; Fig S3 is now Fig 3 and Table S2 is now Table 1 in the manuscript. And, the previous Fig 3 is now Fig S3 and the previous Table S3 is now Table S2 in the supplemental material.

- The addition of the photos in Fig. 2 and Fig. S4 are really helpful, as is the use of the carcass classification scheme to describe their decomposition stage.

***Our response:** Great—we are glad they helpful for the reader.

- Li 278: I would re-phrase this as, “Given that rodents are adept at perceiving and adjusting to risk, and that vegetation did not appear to be a good predictor... , it appears that other features correlated...”

***Our response:** This has been rephrased as suggested in lines 282-285